# SOD1 in ALS: Taking Stock in Pathogenic Mechanisms and the Role of Glial and Muscle Cells

**DOI:** 10.3390/antiox11040614

**Published:** 2022-03-23

**Authors:** Caterina Peggion, Valeria Scalcon, Maria Lina Massimino, Kelly Nies, Raffaele Lopreiato, Maria Pia Rigobello, Alessandro Bertoli

**Affiliations:** 1Department of Biomedical Sciences, University of Padova, 35131 Padova, Italy; catpeg@outlook.it (C.P.); valeria.scalcon@unipd.it (V.S.); k.nies@maastrichtuniversity.nl (K.N.); raffaele.lopreiato@unipd.it (R.L.); 2CNR—Neuroscience Institute, 35131 Padova, Italy; marialina.massimino@cnr.it; 3Department of Radiology, CARIM School for Cardiovascular Diseases, Maastricht University, 6200 MD Maastricht, The Netherlands; 4Padova Neuroscience Center, University of Padova, 35131 Padova, Italy

**Keywords:** amyotrophic lateral sclerosis, SOD1, reactive oxygen species, ROS signaling, Ca^2+^ homeostasis, transgenic mice, skeletal muscle

## Abstract

Amyotrophic lateral sclerosis (ALS) is a fatal neurodegenerative disorder characterized by the loss of motor neurons in the brain and spinal cord. While the exact causes of ALS are still unclear, the discovery that familial cases of ALS are related to mutations in the Cu/Zn superoxide dismutase (SOD1), a key antioxidant enzyme protecting cells from the deleterious effects of superoxide radicals, suggested that alterations in SOD1 functionality and/or aberrant SOD1 aggregation strongly contribute to ALS pathogenesis. A new scenario was opened in which, thanks to the generation of SOD1 related models, different mechanisms crucial for ALS progression were identified. These include excitotoxicity, oxidative stress, mitochondrial dysfunctions, and non-cell autonomous toxicity, also implicating altered Ca^2+^ metabolism. While most of the literature considers motor neurons as primary target of SOD1-mediated effects, here we mainly discuss the effects of SOD1 mutations in non-neuronal cells, such as glial and skeletal muscle cells, in ALS. Attention is given to the altered redox balance and Ca^2+^ homeostasis, two processes that are strictly related with each other. We also provide original data obtained in primary myocytes derived from hSOD1(G93A) transgenic mice, showing perturbed expression of Ca^2+^ transporters that may be responsible for altered mitochondrial Ca^2+^ fluxes. ALS-related SOD1 mutants are also responsible for early alterations of fundamental biological processes in skeletal myocytes that may impinge on skeletal muscle functions and the cross-talk between muscle cells and motor neurons during disease progression.

## 1. ALS and Its Familial Forms: Involvement of SOD1

Amyotrophic lateral sclerosis (ALS) is an invariably fatal neurodegenerative disorder characterized by the selective injury and death of motor neurons (MNs) in the spinal cord, brainstem, and cerebral cortex [1,2]. ALS progressively leads to muscle weakness and atrophy, causing limb motor deficits, dysarthria, dysphagia, dyspnea, and late-stage paralysis, eventually resulting in respiratory failure and death 2–5 years after disease onset. ALS has a worldwide incidence and prevalence that ranges between 0.6 and 3.8 per 100,000 people/year, and 4.1 and 8.4 per 100,000 people, respectively [3], while in Western countries, the incidence is approximately 1–2 per 100,000 people/years and the prevalence is of 5 per 100,000 persons [4]. Comprehensive reviews on ALS incidence and prevalence in different countries are already available [3,5,6,7].

While familial forms of ALS (fALS), mostly involving autosomal dominant forms, are only about 10% of the total, they provide unique opportunities to investigate mechanisms of disease. Several genetic loci have been causally related to both sporadic ALS (sALS) and fALS, with up to 17 genes being definitively associated to ALS (i.e., they statistically increase the risk of disease, https://alsod.ac.uk/, accessed on 13 February 2022). Genes associated to inherited forms include those encoding the TAR DNA-binding protein 43 (TDP-43), the fused in sarcoma/translocated in liposarcoma RNA-binding protein (FUS/TLS), and an intronic GC repeat expansion in the chromosome 9 open reading frame 72 (C9ORF72) [8,9,10]. Other genes have been identified to be related to fALS with a lower frequency and different statistic strength, which—according to gene ontology analysis—cluster in three major categories, RNA processing (e.g., HNRNPA2B1, ATXN2, MATR3), protein trafficking and degradation (e.g., SQSTM1, OPTN, VCP), and cytoskeletal and axonal dynamics (e.g., PFN1D, CTN1) (https://alsod.ac.uk/, accessed on 13 February 2022). Recently, 15 ALS-risk loci have been identified in a genome-wide association study by combining new and pre-existing datasets from ALS and control patients, supporting the complex genetics behind ALS, as well as the relationship between genetics and other individual characters (i.e., clinical, environmental, behavioral) [11].

The first gene whose mutations were linked to fALS, however, was that encoding the cytosolic enzyme Cu/Zn superoxide dismutase 1 (SOD1), a redox protein able to convert superoxide anion to hydrogen peroxide (H_2_O_2_). Following the seminal linkage analysis published almost 30 years ago [12], other studies reported several ALS-associated SOD1 mutations, leading to the notion that they account for about 3% of total ALS cases (20% of fALS and 2–7% of sALS), with more than 180 mutations discovered thus far (https://alsod.ac.uk/, accessed on 13 February 2022). As for the frequency of the different SOD1 mutations in ALS, data have been reported, for example, in the Chinese or Italian populations by previous analyses [13,14].

The relative prevalence of the main gene mutations that have been associated with ALS is given in Figure 1.

Thanks to such remarkable genetic advancements, different animal (rodent) models have been developed over time, expressing ALS-related mutants of the above genes. Some shared pathways leading to cell damage—including protein misfolding and aggregation, oxidative stress, glutamate excitotoxicity, disturbed Ca^2+^ homeostasis, endoplasmic reticulum and mitochondrial dysfunctions, defective RNA editing and control of protein synthesis, neuroinflammation, and loss of neurotrophic factors—have been proposed as pathogenic mechanisms in patients and animal paradigms. Unfortunately, however, all these proteins/genes were found to have divergent roles and generate different ALS phenotypes, and thus a comprehensive view is still lacking. Among such animal models, those based on ALS-related SOD1 mutants are the most exploited in the field. In this respect, cell models, ranging from the simple unicellular organism yeast to human-derived induced pluripotent stem cells (iPSCs), are also being continuously exploited to uncover fundamental pathogenic mechanisms in SOD1-related ALS forms and other diseases.

Despite such a richness of experimental paradigms, however, it has still to be definitively clarified as to whether SOD1-related ALS forms were initiated by a detrimental loss of activity of the enzyme, or acquisition of novel neurotoxic properties of the protein—possibly caused by cell intoxication by misfolded and aggregated protein conformers—or both mechanisms. Such harmful events specifically target MNs, albeit a direct or indirect involvement of other cell types (e.g., glial and/or skeletal muscle cells) has been suggested by a wealth of evidence.

The aim of the present review is to take stock of some of the latest findings on SOD1 involvement in ALS. Since comprehensive reviews have been recently published on this subject (to name a few, [5,15,16]), we here particularly focus on some aspects of the complex relation between SOD1 and ALS, and the current research thereof, with a minor attention to MNs for which readers may refer to the above comprehensive reviews. Topics include the relationship between the altered SOD1 structure and its dysfunction in ALS, SOD1-based ALS models, the involvement of oxidative stress in disease etiology, and the role of non-neuronal cell types in pathogenesis. Specific attention will be given to recent research on skeletal muscle cells and alterations of Ca^2+^ homeostasis as possible players in the SOD1–ALS connection, also providing novel data on Ca^2+^ alterations in human (h) SOD1(G93A) mouse primary myocytes.

## 2. SOD1 in ALS Pathogenesis: SOD1 Structure and Function in Relation to ALS

SOD1 is one of the most abundant proteins ubiquitously expressed in mammalian tissues, with the highest levels being detected in liver and in the central nervous system. SOD1 is a homodimeric metalloenzyme, in which the two subunits of 16 kDa (153 amino acids in the human isoform) are connected by a disulfide bridge [17]. Each subunit is formed by a β-barrel core and seven loops at the edge stabilized by an intramolecular disulfide bond, a binuclear metal binding site, and a series of hydrogen bonds. The metal binding site embraces a copper and a zinc ion that make SOD1 fully functional in its catalytic activity by binding one Zn and one Cu ions per monomer (Figure 2) [17,18,19,20,21,22].

Notably, SOD1 structure is evolutionarily conserved from prokaryotes to humans, pointing to the relevance for all living cells of SOD1 enzymatic activity, i.e., the conversion of superoxide anions into molecular oxygen and H_2_O_2_. The simple eukaryotic model organism *Saccharomyces cerevisiae* has been exploited to scrutinize fundamental aspects of the SOD1 structure/function relationship, by the analysis of the yeast paralogue Sod1p, sharing with human SOD1 either structural (70% overall homology, and more than 90% identity in the active-site) and enzymatic (Cu/Zn-dependent activity) features, and intracellular distribution, characterized by a diffused presence in the cytosol and the nucleus [23].

As for higher organisms, in yeast cells the role of SODs is also mainly linked to the protection from reactive oxygen species (ROS) damage (O^2−^ detoxification). However, the mutation of SOD genes may affect multiple cellular processes: besides higher sensitivity to oxidative stress [24,25,26], yeast SOD mutants display alterations in either copper homeostasis, amino acids metabolism, and mitochondrial functions, as well as the reduction of both replicative and chronological life span (longevity/ageing) [24,27,28,29,30,31]. Further investigations in *S. cerevisiae* unveiled the crucial implication of Sod1p in cell physiology, with primary roles in the response to environmental challenges, such as the regulation of cellular respiratory rate, linked to nutrient and oxygen availability [32,33,34]. Additional functions have been reported, indicating the ability of Sod1p to act as a transcription factor (TF) upon its nuclear translocation [35,36] and—recently—yeast Sod1p has also been implicated in the regulation of the cellular redox state, directly affecting protein oxidation by H_2_O_2_, a product of its catalytic action [37,38].

Interestingly, SOD1 is noticeably stable, also to pH changes and relatively high temperature, and maintains the intrasubunit disulfide bond even in reducing conditions [22]. Such quaternary structural integrity of holo-SOD1 is maintained by Cu/Zn metalation and the presence of intrasubunit disulfide bonds between Cys57 and Cys146, and either the loss of the Cu/Zn or the disruption of the disulfide bonds may result in pathogenic misfolding [39,40]. It is worth noting that the status of the intrasubunit disulfide bonds and the molecule metalation are able to reciprocally influence each other. Indeed, the breaking of the disulfide linkage can cause metal loss [41], indicating that the protein disulfide bond controls the Cu/Zn coordination [42], while, reciprocally, the correct coordination of Cu/Zn to SOD1 is critical for the formation of disulfide bonds and the maintenance of the enzyme structural integrity [22].

Copper appears to play a dominant role in the kinetic stability of SOD1 protein, and Cu insertion depends on the activity of the copper chaperone Ccs1 [43]. Notably, the yeast orthologue Ccs1p also plays a further regulatory role in Sod1p localization, which, under oxidative stress conditions, is partially accumulated in the mitochondrial inter-membrane space where it can contribute to reduce ROS levels [44,45].

In addition, the coordination of the Zn ion drives the local folding of two disordered loops into a catalytic subdomain and stabilizes SOD1 structure [46].

Recent data indicate a crucial role played by Cu in ALS pathogenesis. Indeed, in the central nervous system of hSOD1(G37R) mice, an insufficient bioavailability of copper was reported, so that—unable to counterbalance the increased activity of the over-expressed protein—the missing metalation of SOD1 causes the protein misfolding and aggregation [47].

SOD1 activity is also regulated by other post-translational modifications. The most important ones are the phosphorylation at Ser60 and Ser99 residues, triggering SOD1 nuclear localization, and the ubiquitination of Lys63, regulating its degradation, while acetylation (at Lys70, Lys122), phosphorylation (at Thr39), glycation (at Lys122, Lys128), and succinylation (at Lys122) may represent key events in SOD1 maturation and its antioxidant activity [48].

Recently, it has been demonstrated in yeast and in mammalian cells that SOD1 rapidly enters the nucleus in response to increased level of H_2_O_2_. Indeed, ROS promotes SOD1 association with the Mec1/ATM effector Dun1/Cds1 kinase that phosphorylates SOD1 at Ser60 and Ser99, allowing it to shuttle to the nucleus where it acts as a TF, regulating the expression of genes involved in oxidative stress resistance and preventing oxidative genomic damage [35].

However, phosphorylation of other residues modulates SOD1 activity in different manners. For example, phosphorylation at Thr39 or Thr40 (in yeast and human, respectively) by mTORC1 regulates SOD1 activity by coupling nutrient availability with ROS level [33].

In addition, SOD1 stabilization and regulation is also affected by cysteine redox modulation. Indeed, hSOD1 presents four cysteine residues (Cys6, Cys57, Cys111, and Cys146) in each subunit. As mentioned before, Cys57 and Cys146 form an intrasubunit disulfide bond that maintains the structure and catalytic activity, whereas Cys6 and Cys111 are present in a reduced state with free sulfhydryl groups and are in the core of each subunit and on the protein surface, respectively. Cysteines are susceptible to oxidative modification, and Cys111 can undergo to oxidation or glutathionylation, determining the dissociation of the SOD1 functional dimer [48,49].

Changes in the redox state of cysteine residues have also been found in ALS-linked mutant SOD1, which shows more susceptibility to reducing conditions driving to intrasubunit disulfide bond reduction. Such an event causes dimer dissociation and the exposition of the SOD1 hydrophobic region that can therefore interact with other subunits leading the formation of protein aggregates [50].

## 3. SOD1 Mutations and ALS Models

While it is well established that a causal relation exists between SOD1 and ALS, the mechanisms of SOD1-related neuropathology are far from being clearly understood. A long-lasting debate concerns the balance between the enzyme loss of function and its gain of toxic capacity in driving disease phenotypes. To discern such a correlation, different SOD1 models were generated, which will be discussed hereafter.

### 3.1. SOD1(G93A) Rodent ALS Models

Soon after the seminal discovery of SOD1-related fALS cases, the first transgenic (Tg) mouse model was developed, carrying multiple copies of the hSOD1(G93A) transgene over a mixed genetic background that closely resembled several traits of human ALS [51]. Such mice, expressing high levels of hSOD1 with the ALS-associated substitution of an alanine for a glycine at position 93, suffered from a severe MN disease, leading to serious motor impairment by around 120 days of age and becoming terminally ill at around 160 days. Disease was age-dependent and progressive, with early slowing of growth and appearance of tremors and shortening of stride, followed by muscle weakness and atrophy and eventually paralysis [51,52]. Histopathological analyses revealed that such striking phenotypes were accompanied by progressive neuropathology, including early neuronal vacuolation and subsequent loss of MNs in the anterior horns of the spinal cord, loss of myelinated axons, and muscle denervation and reinnervation [52,53,54]. Interestingly, the high levels of expressed (mutant) SOD1 were also paralleled by high enzymatic activity, which was similar to that observed in mice expressing comparable amounts of wild-type (WT) hSOD1 that did not develop any clinical sign of disease [52,54]. In addition, it soon became clear that disease onset and lifespan were strongly dependent on the number of transgene copies, given that animals expressing lower levels of hSOD1(G93A) over the same genetic background developed disease much later [54]. According to the above observations, Gurney and colleagues ended their pioneering work by suggesting that “dominant, gain-of-function mutations in SOD play a key role in the pathogenesis of familial ALS” [51].

Such a high-expressing hSOD1(G93A) mouse model soon became the most employed animal paradigm for the study of ALS and—in spite of some drawbacks, including the fact that disease severity was rather variable over time and from lab-to-lab (possibly reflecting transgene copy number variations over repeated breeding), as well as the fact that treatments successfully alleviating disease phenotypes in these mice failed in clinical trials [55,56]—it is still widely diffused in research and accepted by most of the ALS scientific community nowadays.

While the G93A mutation is relatively rare among all SOD1 ALS forms, other hSOD1(G93A) transgenic mouse strains have been created over time, providing strong evidence that disease duration and survival in these animal models were also remarkably dependent on the genetic background [57,58,59]. The first evidence with this respect was the observation that breeding the original Tg strain with C57B6/SJL mice significantly reduced survival time [60]. The influence of the genetic background on the disease course was then replicated by different laboratories (see [57] for a review), and a dedicated study eventually demonstrated that C57BL/6JOlaHsd mice had strikingly delayed disease onset and slowed progression compared with a 129SvHsd strain, despite comparable levels of expressed hSOD1(G93A) transgene [61,62]. Importantly, such differences were accompanied by different neuropathology and cellular and biochemical signatures [57,62], possibly reflecting the phenotypical and clinical variability reported in humans.

### 3.2. Other SOD1 Mutations in ALS

Following the seminal discovery of the G93A mutation in ALS patients [12], more than 100 SOD1 genetic variants have been found associated to ALS (https://alsod.ac.uk/, accessed on 13 February 2022). Most of them have been found in fALS, but some mutations are also related to sALS. SOD1 fALS forms are prevalently inherited with dominant transmission (but also recessive mutations have been identified) and present with a heterogeneity of pathology (e.g., differential involvement of upper and lower MNs) and clinical manifestations depending on the type of the mutation and other genetic or environmental factors. For example, the A4V mutation, the most common fALS SOD1 mutation among the North American population, results in a very aggressive phenotype with early onset and rapid progression, with death occurring about one year after the clinical manifestation [5,63]. D90A is the most prevalent mutation in Europe, showing either a dominant or recessive pattern of inheritance depending on the population [64]. This mutation is generally associated to a milder disease phenotype compared to the A4V SOD1 variant, with slower progression and respiratory failure occurring in 3–5 years. In some specific populations (i.e., from Northern Europe), survival may also prolong up to 10 years after diagnosis, suggesting a linkage to other genetic traits. In this case, classical motor phenotypes are often accompanied by cognitive deficits, possibly reflecting the late stage spread of disease to nonmotor neurons [5,63,65,66].

Such genetical insights and the observation that the clinical courses of sALS and fALS were similar in several aspects prompted the production of Tg mouse models bearing SOD1 mutations different from hSOD1(G93A) (and of other fALS-related genes) for the study of ALS pathomechanisms [67,68,69]. Such Tg mouse models included the G37R, G85R, G86R, and D90A SOD1 missense mutants [51,70,71,72], which—although mimicking human traits of disease—strongly supported the notion that different mechanisms contribute to ALS pathogenesis [65,67].

### 3.3. Other Models of SOD1-Related ALS

The yet unclear mechanisms of SOD1-related pathology have also been investigated in simple model organisms (*C. elegans*, *D. melanogaster*, *D. rerio*, mammalian primary cell cultures and cell lines, and human-derived iPSCs) [69,73], including yeast cells that can be suited for understanding basic molecular pathways involved in SOD1 toxicity. Indeed, thanks to the above-described Sod1p properties and the ability of hSOD1 to functionally replace the yeast enzyme [74], *S. cerevisiae* has been used as a model organism to investigate the effect of fALS related SOD mutations. Multiple genetic, functional, and biochemical assays have provided evidence that fALS-linked mutations may have a profound impact on the protein stability and the formation of insoluble aggregates, correlating protein misfolding and cytotoxicity [75,76,77]. Further studies in yeast models supported the notion that fALS-SOD1 mutations are causative of mitochondrial and respiratory defects [78], but also indicated that SOD1-dependent toxicity may be correlated to metabolic dysfunctions, including the regulation of amino acid metabolism [79]. Recently, the relationship between cell toxicity and enzyme maturation has been also investigated by characterizing either assembly, stability, and properties of heterodimeric complexes composed of WT and fALS mutant hSOD1 isoforms (i.e., A4V, L38V, G93A, G93C) [80]. The overall experimental evidence indicated that the association of WT/mutant hSOD1 heterodimers, in both yeast and human cell models, promoted protein misfolding and the formation of insoluble aggregates, thus increasing cytotoxicity. Consistently, the levels of intracellular antioxidant activity were impaired, being particularly critical for yeast cells under aging conditions, where it could be exacerbated the oxidative stress-induced mutant SOD1 misfolding.

## 4. SOD1 in ALS Pathogenesis: Mechanisms and Non-Neuronal Cells Involved in ALS

As discussed previously, SOD1-related ALS pathomechanisms may involve disturbances of basal cell functions in MN-interacting cell types, such as altered redox balance and Ca^2+^ dyshomeostasis in both glial and skeletal muscle cells. Reviewing of these aspects of SOD1/ALS relationships is reported below.

### 4.1. Altered MN-Glial Communication and Oxidative Stress in ALS

Oxidative stress depends on the excessive production of ROS and reactive nitrogen species, which cannot be efficiently detoxified by the cellular antioxidant systems. Overproduction of reactive species, along with failure of the balance maintained by the antioxidant enzymatic systems, results in the damage of the major cellular constituents such as lipids, proteins, and nucleic acids [81]. Neurons are particularly susceptible to a redox imbalance because of their high metabolic requirements and large size [82]. Of note, low levels of reduced glutathione (GSH) in the motor cortex [83], a systemic pro-inflammatory state (i.e., increased levels of IL-6 and IL-8), and impaired antioxidant systems [84] have been found in ALS patients. However, although oxidative stress has been repeatedly implicated in ALS pathology, it is still unclear as to whether it is a causative event or a downstream effect of disease progression [85].

Of note, neurons heavily rely on glial cells for their survival. Neuroglia comprises both the microglia formed by cells of the immune system and the macroglia, which is principally constituted by astrocytes. Previous findings highlighted that astrocytes expressing mutated SOD1 are toxic to co-cultured WT MNs [86]. Accordingly, astrocytes derived from autopsy samples from sporadic ALS patients are also toxic to MNs [87], while, conversely, WT glial cells are able to protect neurons expressing mutant SOD1 [88]. Thus, an altered glia-to-neuron communication could be important in ALS pathogenesis.

Besides cytokines, growth factors, and gliotransmitters, glial cells can secrete a variety of proteins that can play a relevant role in intercellular communication. These proteins can be free-circulating or secreted inside vesicular systems. Some antioxidant enzymes are already known to be released into the cerebrospinal fluid [89]. In particular, thioredoxin 1 (Trx1) is a recognized neurotrophic factor supporting nerve growth factor signaling for neurite outgrowth [90]. Interestingly, Trx1 secretion can be affected by oxidative stress conditions [91], and altered levels of Trx1 have also been reported in other neurodegenerative disorders, such as Alzheimer’s disease [89]. Furthermore, Trx1 was found to be highly upregulated in erythrocytes from patients with familial ALS bearing the G37R or H46R SOD1 mutations [92]. Trx1 is a small component of the thioredoxin system, together with NADPH and the selenoenzyme thioredoxin reductase (TrxR). It is now critical to determine whether an alteration of the redox balance is generating the cascade of events leading to protein aggregation and neuronal cell death, and whether it is possible to counteract this process by modulating the cellular redox signaling from glia towards neuronal cells.

Notably, proteomics and metabolomics data obtained from primary cultures of hSOD1(G93A) astrocytes provided evidence of reduced expression of proteins and metabolites implicated in the GSH metabolism, supporting the hypothesis that an unbalanced redox signaling is implicated in ALS glia–MN connection [93].

The relevance of oxidative stress in ALS pathogenesis is also corroborated by the fact that the redox unbalance in hSOD1 Tg mice causes an aberrant disulfide bond formation between the normally unpaired Cys6 and Cys111, leading to SOD1 misfolding and consequent aggregation [94]. Interestingly, elevated intracellular ROS also causes aberrant disulfide cross-linking in other ALS-related proteins, such as TDP-43 and FUS/TLS [95,96].

As the properties of mutant SOD1 that determine toxicity remain not completely defined, many therapeutic approaches are proposed in different directions going from gene therapy in mutant SOD1-associated fALS, including RNA interference (RNAi), SOD1 antisense oligonucleotides [97,98], and CRISPR/Cas9 system targeting the mutant SOD1 gene [99], to other strategies such as those involving, for example, small molecules, peptides, and monoclonal antibodies [100].

Considering that in sALS and fALS patients, alterations in anti-oxidative enzymes in brain, spinal cord, CSF, and blood specimens were found and that SOD1 is an anti-oxidant enzyme [101], the latter approach has, however, mainly focused on antioxidant molecules. Preclinical studies of antioxidants as potential therapeutic agents have showed promising results, but despite the positive outcomes in neuronal cell cultures and in mouse models, the same positive effects were not replicated in patients. Edavarone, a free-radical scavenger and potent antioxidant, was one of the major molecular drugs proposed for clinical trials in ALS patients. This drug was first indicated in the treatment of acute ischemic stroke [102], and then it was inserted in a phase III clinical trial, showing a slowing of ALS progression [103]. However, many other studies are required, because a multicenter study in patients with ALS did not lead to positive outcomes [104].

Rasagiline, a monoamine oxidase B inhibitor, was also studied for its possible effects on ALS, but it did not show alteration of disease progression in a randomized controlled trial [105].

Riluzole, a benzothiazole with antiglutamatergic properties that shows also antioxidant features, is the only established disease-modifying drug for ALS, although its efficacy in patients is very mild and disease stage-dependent [106,107], while it has been found unable to provide any significant benefit on lifespan and motor performances in SOD1(G93A) mice [108].

Many other antioxidants molecules have been proposed, including vitamin E [109], N-acetyl-L-cysteine, coenzyme Q10, epigallocatechin gallate, curcumin, and Nrf2/ARE modulators, but unfortunately with uncertain results and are thus still under investigation (for further details on this topic, see the reference [110]).

### 4.2. SOD1 in ALS Skeletal Muscle

As anticipated, the major target of ALS neuro-muscular pathology is MNs. However, skeletal muscle is also involved—and may play a major role—in disease pathogenesis. There is no doubt that dysfunctions of skeletal myocytes (SMs) arise from MN death and dismantlement of the neuromuscular junction (NMJ). However, it has also been proposed that under some circumstances, primary skeletal muscle damage may prime, or contribute to, MN demise through a sort of dying back mechanism [111], particularly in the case of SOD1-related ALS forms [112,113].

In the diagnosis for ALS, muscle biopsies were rarely used, and therefore histopathological analysis throughout the course of disease are not available, with postmortem investigations revealing skeletal muscle damage only in the final stages. The very few older studies on muscle biopsies from ALS patients, based on morphometric and histochemical studies, showed atrophic fibers and a process of denervation and reinnervation evidenced by increased terminal innervation of denervated fiber by collateral sprouting of axons from healthy MNs [114].

Given these assumptions, the use of Tg mouse models, especially hSOD1(G93A) mice, was valuable in the study of skeletal muscle in fALS because it allows a complete spatiotemporal analysis of the tissue during pathology progression.

Several studies reported skeletal muscle alterations in SOD1-related fALS animal models, such as the presence of SOD1-containing aggregates and ROS formation/accumulation, impaired skeletal muscle regeneration, disappearance of NMJs, and muscle atrophy. Importantly, skeletal muscle alterations occur before MN death and the onset of clinical symptoms [111,115].

Turner and colleagues found aggregates in the gastrocnemius of hSOD1(G93A) mice and hypothesized that they could contribute to hind limb degeneration and paralysis [116]. In contrast, a subsequent paper showed muscle degeneration associated with mitochondrial dysfunctions and increased ROS release in the absence of hSOD1(G93A) aggregates [117]. An exacerbated accumulation of ROS was found in the skeletal muscle of hSOD1(G86R) mice [72], and an upregulation of SOD1 activity was also observed in muscles of mice carrying the G93A mutation during the progression of disease, indicating the involvement of oxidative stress [118,119]. Skeletal muscles of hSOD1(G93A) mice also displayed increased oxidative stress with elevated formation of mitochondrial ROS, leading to mitochondrial dysfunctions during disease progression [120].

Skeletal muscle possesses a high regenerative capability that, however, has been shown to be impaired by oxidative intracellular environment because ROS overload interferes with myogenic differentiation [121,122]. Intriguingly, expression of hSOD1(G93A) in the C2C12 immortalized muscle cell line induces alteration of myogenesis by inhibiting differentiation and directing myocytes toward an adipogenic phenotype [123]. In addition, the myogenic precursor satellite cells from hSOD1(G93A) Tg mice showed impaired proliferation during all stages of disease progression, suggesting reduced myogenic potentials [124].

Early tremor and weakness observed in hSOD1(G93A) mice [51] prompted the analysis of NMJs, and all studies in this regard reported selective loss of motor units in fast-twitch muscles (starting from ≈50 days before symptom onset).

In both ALS patients and mouse models of mutated SOD1, damage to NMJs seems to occur at early disease stages, before massive MN degeneration and appearance of symptoms [111] and increases as the disease progresses causing loss of contractile force [125,126,127]. These results support the idea that MN death is not a totally cell autonomous event and that the damage to muscle-terminal nerve synapses may concur to the neurodegenerative phenotype.

To demonstrate that skeletal muscle is a primary target of mutant SOD1 toxicity, two Tg mouse models were created in which mutated or WT SOD1 is specifically expressed in such a tissue only [128,129].

In the first model, mice carried the hSOD1(G93A) mutant under the control of the skeletal muscle-specific promoter from rat myosin light chain (MLC)-1, resulting in higher expression of the MLC/SOD1(G93A) transgene in muscles enriched in fast fibers (the most damaged in ALS), and lower in slow muscles such as the soleus, of adult mice [128]. This selective expression of hSOD1(G93A) generated skeletal muscle accumulation of ROS and atrophy, resulting in reduced strength, sarcomeric disorganization, and alterations in mitochondrial morphology. In subsequent papers, the same authors showed that the muscle phenotype in MLC/hSOD1(G93A) mice was due to accumulation of ROS, which in turn alters the Akt pathway, reducing protein synthesis and activating the FoxO pathway, promoting protein breakdown and muscle atrophy. One of the most striking observations in this animal model was that muscle atrophy is an early event that occurs before MN degeneration. Early atrophy is followed by NMJ degeneration, spinal cord astrocytosis, and subsequent apoptotic MN death that in turn worsens muscle atrophy and induces a fast-to-slow shift in fiber type composition [112].

The change in muscle fiber types from glycolytic to oxidative was observed in muscles of both ALS patients and mice expressing hSOD1(G93A) or (G86R) [114,126,127,130,131,132], as well as in MLC/hSOD1(G93A) muscles where an increase in lipid catabolism was also found [133]. Moreover, the downregulation of L-type Ca^2+^ channel (CaV1.1, a positive regulator of muscle mass) contributes to the atrophic process of MLC/hSOD1(G93A) muscles [134]. In support of the hypothesis that muscle can positively or negatively influences the nervous system, results obtained from miRNome and transcriptome analysis of the spinal cord MLC/hSOD1(G93A) showed alteration of genes involved in myelin processing [135].

In the second mouse model, the specific over-expression of WT or mutated (G37R or G93A) hSOD1 in the skeletal muscle is driven by the chicken α_sk_ actin promoter [129]. In this paradigm, hSOD1(WT) led to the same pathological phenotype of the two ALS-associated mutants [129]. The toxicity is caused by oxidative stress, rather than by protein aggregates, as cells exhibit oxidative damage and protein nitration, and also in these Tg mice, the pathological process starts from the muscle to subsequently damage the NMJ, leading to MN death in the spinal cord [113].

The results obtained in both mouse models suggested that skeletal muscle dysfunctions in SOD1-related ALS may also be caused by direct muscle cell toxicity of over-expressed/mutated SOD1 and are not just a consequence of denervation associated with MN death, proposing that a non-autonomous process driven by skeletal muscle contributes to MN degeneration in ALS.

Increasing evidence suggest that two different mechanisms may play a role in the altered MN/skeletal muscle interplay in SOD1-related ALS. The first one is the unbalanced abundance/activity of two NAD+ dependent protein deacetylases belonging to the Sir2 family protein (SIRT1/3) [136]. Indeed, SIRT3 was implicated in muscle metabolic changes in SOD1(G93A) mice [137], while the NMJ degeneration observed in SOD1(G93A) mice was related to an altered expression of SIRT1 [138]. Moreover, several studies in the SOD1(G93A) mouse model, using SIRT1 activators such as resveratrol, have shown an improvement in survival and slowing disease progression [139,140,141]. Taken together, these findings suggest SIRT1/3 as possible promising therapeutic targets for SOD1-related ALS [137,142].

The second mechanism involves Ca^2+^ dyshomeostasis and will be discussed in a more detailed way in the next paragraph.

### 4.3. Disturbances of Ca^2+^ Homeostasis in SOD1-Related ALS

The pathophysiological processes linked to ALS neurodegeneration—such as the increased oxidative stress and free radical damage, mitochondrial dysfunction, and glutamate excitotoxicity—own a key factor, that is, the disturbance of Ca^2+^ homeostasis in MNs [143,144,145,146,147,148], as well as in astrocytes [149,150] and skeletal muscle cells [151].

Such perturbations have been well-documented in the hSOD1(G93A) Tg mouse, in which the Ca^2+^ overload in the cytoplasm [143,144,145,148], the impairment of mitochondrial Ca^2+^ capacity [152,153,154], and alterations of the mitochondrial Ca^2+^ extrusion system [154,155] have been reported. In addition, an increased intracellular Ca^2+^ level was also reported in the spinal cord and skeletal muscle of sALS/fALS patients [156,157,158], in other ALS-related Tg mouse models [159], and in MNs derived from iPSC cell lines of fALS patients [160,161]. This suggests that the lack of a finely tuned Ca^2+^ homeostasis plays a pivotal role in ALS disease, although mechanisms leading to Ca^2+^ disturbance may be specific to each ALS-related mutation. Several reports indicate that such alterations may be caused by the modified expression/functionality of proteins involved in Ca^2+^ homeostasis. Indeed, an intriguing series of studies suggested that SOD1 mutant MNs are characterized by altered expression of the sodium-calcium exchanger (NCX) 1 and 2 [162] and of the Ca^2+^ permeable AMPA receptors, that is also characterized by a higher relative Ca^2+^ permeability ratio [163,164]. Moreover, several studies demonstrated that altered gene editing and/or perturbed expression of the GluR2 AMPA subunit (important to reduce the receptor Ca^2+^ permeability) causes excitoxicity, dramatically accelerating MN degeneration in ALS [148,165,166,167].

Altered localization and function of the voltage-gated Ca^2+^ channels were unveiled in cultured SOD1(G93A) MNs, causing increased high voltage-activated Ca^2+^ currents that may contribute to ALS onset and progression [168]. Interestingly, an elevated expression of metabotropic glutamate receptors was detected in iPSC-derived MNs from SOD1(G93A) fALS patients, causing a reduced intracellular Ca^2+^ storage release [160]. Moreover, a subnormal level of the Ca^2+^ buffering proteins parvalbumin and calbindin was previously detected in MNs derived from hSOD1(G93A) Tg mice [169], which was subsequently confirmed in MN axons of ALS patients [170], thus suggesting a reduced Ca^2+^ buffering capacity in pathological conditions. Recently, a RNAseq study unveiled low levels of miR-18b-5p in different SOD1 models (NCS34 cells expressing different SOD1 mutants, SOD1(G93A) Tg mice and in fALS (G86S and G17S) patients). Such alterations are linked to reduced expression of Mtcp1, a protein essential to neuronal Ca^2+^ signaling, causing increased intracellular levels of the ion [171].

It is of paramount importance to emphasize the fact that disturbances of Ca^2+^ metabolism in ALS may involve multiple cell types besides MNs. In this context, an impaired Ca^2+^-dependent astrocyte-MN cross-talk strongly contributes to ALS pathogenesis. The first evidence of such a correlation was demonstrated in the cerebrospinal fluid of fALS patients, and also in SOD1(G93A) and (G37R) Tg mice. In this case, a reduced expression of the glial glutamate transporter (EAAT2) by astrocytes, essential for glutamate clearance from the synaptic cleft, represented the cause of glutamatergic overstimulation and Ca^2+^ overload in ALS-affected MNs [70,172,173]. Moreover, several studies revealed pathogenic changes in ALS astrocytes, including disrupted Ca^2+^ signaling [149,150,174,175] and altered expression/activation of proteins involved in Ca^2+^ homeostasis [149,150] that may contribute to the astrocyte-dependent toxicity to MNs.

In this respect, in a recent paper, we reported altered patterns of protein expression and TF activation in hSOD1(G93A) astrocytes [93], some of which (i.e., RXRA, CTCF, and NF-kB) have been previously recognized to regulate proteins related to Ca^2+^ homeostasis, and may also correlate with altered Ca^2+^ mobilization in glial cells [149,150]. In more detail, RXRA regulates Ca^2+^ release from the ER and mitochondrial Ca^2+^ uptake through the transcriptional repression of the inositol 1,4,5-trisphosphate receptor type 2 (ITPR2) and the regulation of the mitochondrial Ca^2+^ uniporter (MCU) activity, which may in turn control mitochondrial ROS production and ROS-dependent DNA damage and cellular senescence [176]. CTCF, a major regulator of chromatin structure in mammals, controls the splicing and expression of voltage-dependent Ca^2+^ channels [177]. Intriguingly, such an activity may also speculatively be reminiscent of the (post-transcriptional) regulation of pre-mRNAs typical of the ALS-associated TDP-43 protein [178]. Finally, it has been well known for years that the NF-κB TF regulates the expression of proteins involved in Ca^2+^ transport and Ca^2+^-dependent cellular functions, such as store-operated Ca^2+^ entry [179], that has been previously associated by us and others to neurodegenerative disorders [180,181,182,183].

This ensemble of evidence, pointing to an important involvement of Ca^2+^ homeostasis deregulation in SOD1-related ALS, prompted us to analyze some aspects of Ca^2+^ metabolism in SMs from the SOD1(G93A) Tg model. Indeed, as discussed previously, skeletal muscle fibers also seem to play an active role in ALS pathogenesis. Nevertheless, the existence of altered Ca^2+^ fluxes in ALS skeletal muscle has only been poorly investigated [151]. With the aim of identifying aberrancies in Ca^2+^ homeostasis in SMs, we recently investigated possible alterations of Ca^2+^ regulation mechanisms in hSOD1(G93A) SMs by analyzing the expression of Ca^2+^ transporting systems and local Ca^2+^ movements in primary cultured cells differentiated in vitro to the stage of myotubes.

Following the notion that a synergy between mitochondrial Ca^2+^ alterations and ROS production may play a relevant role in ALS [153,184,185], as well as the reported Ca^2+^-related proteome modifications in SOD1(G93A) cells, we firstly checked the expression of mitochondrial and sarcoplasmic reticulum (SR) Ca^2+^ transporters in primary SMs from hSOD1(G93A) Tg mice compared to the hSOD1(WT) counterpart as control.

Such analysis showed a remarkable decrease in the sarco/endoplasmic reticulum Ca^2+^ ATPase isoforms SERCA1 and SERCA2, mediating SR Ca^2+^ uptake and store refilling; ryanodine receptor (RyR), a regulating agonist induced release of the ion from the SR; and the channel subunit of MCU, mediating mitochondrial Ca^2+^ uptake [186], in hSOD1(G93A) SMs (≈40%, ≈40%, ≈50%, and ≈60% reduction for SERCA1, SERCA2, RyR, and MCU, respectively). Conversely, the plasma membrane Na^+^/Ca^2+^ exchanger (NCX) and plasma membrane Ca^2+^ ATPase (PMCA) did not significantly change, suggesting selectivity for mutant SOD1 to modify the (transcriptional) regulation of Ca^2+^ transporting proteins (Figure 3).

Starting from such results, we measured SR-mediated mitochondrial Ca^2+^ entry by transfecting primary cultures of SMs from control and hSOD1(G93A) Tg mice with a plasmid encoding aequorin, a genetically encoded Ca^2+^ sensitive photo-protein suited for cell-population assessments of Ca^2+^ fluctuations, specifically targeted to the mitochondrial matrix (mitAEQ, [188]). Cells were stimulated with caffeine, which induces the release of Ca^2+^ from the SR by RyR activation. Consequently, Ca^2+^ release causes both an increase of cytosolic [Ca^2+^] and of Ca^2+^ uptake into mitochondria located in close proximity to the SR [189]. Figure 4a,b shows that in vitro differentiated hSOD1(G93A) myotubes have a significantly lower (≈20%) mitochondrial Ca^2+^ uptake compared to healthy controls. This event is reflected by a parallel decrease (≈12%) of the Ca^2+^ transient in the cytosol, monitored by transfecting SMs with a plasmid coding for a cytosolic aequorin isoform (cytAEQ, [188]) (Figure 4c,d).

The difference in the mitochondrial [Ca^2+^] transient during stimulation with caffeine is likely a combination of both reduced SR release and mitochondrial uptake. The first event may be justified on the basis of both altered expression of the SERCA pumps (regulating the uptake of Ca^2+^ in the SR) and/or reduced RyR activity, while the second occurrence may be related to the impaired capacity of mitochondria to drain SR-released Ca^2+^ due to the lower MCU expression. Unfortunately, however, we were unable to measure ER basal Ca^2+^ levels, which would have definitively supported this notion, due to technical constraints.

Albeit such results deserve more investigations and are far away to set a definitive conclusion on the mechanisms linking mitochondrial Ca^2+^ to ALS, yet they provide an advancement in the complex relationship between mitochondrial Ca^2+^ handling and ROS production in disease pathogenesis. Indeed, it is well established that Ca^2+^ and ROS are linked by a mutual interplay useful to strictly regulate cellular signaling networks. Interestingly, such connection links Ca^2+^ and ROS metabolism with each other [153]. For example, Ca^2+^ modulates several systems/enzymes involved in ROS production, in such a way that elevated intracellular Ca^2+^ levels activate ROS-generating enzymes and free radical formation. In a similar way, redox state and ROS are involved in the modulation of the activity of a variety of Ca^2+^ channels, pumps, and exchangers, by primarily controlling their oxidative state.

Increasing evidence suggests that an alteration of this finely tuned cross-talk may be responsible for many disorders, including ALS among other neurodegenerative diseases [153,192]. In particular, many studies, accomplished in different ALS models (mutant SOD1, TDP-43, or FUS/TLS), demonstrated that an impairment of the ER–mitochondria communication [188,193,194] causes an altered mitochondrial Ca^2+^ capacity, inducing a significant decrease in the mitochondrial membrane potential and an increase in the rate of the production of mitochondrial ROS [153]. However, further findings are needed to finally determine if the dysfunctional Ca^2+^ fluxes observed in ALS-related primary SMs—that we may argue be related to an altered expression of Ca^2+^ transporters—lead to an unbalanced redox signaling and impaired SM functionality.

Our observation of decreased transient Ca^2+^ movements in the cytosol and mitochondria upon caffeine stimulation may appear to contradict previous reports showing increased intracellular Ca^2+^ levels in SOD1(G93A) cells [144,156,195]. However, it should be considered that the response of Ca^2+^ transient oscillations is strictly dependent on the kind of stimuli. For example, in a recent paper, we reported increased Ca^2+^ entry from the outside (and higher cytosolic and mitochondrial Ca^2+^ peaks) following stimulation of store-operated Ca^2+^ entry in primary SOD1(G93A) astrocytes, which may well be consistent with the reduced SERCA level that we observed in that cell type [150].

Further, one has to distinguish between Ca^2+^ fluxes upon cell stimulation and basal Ca^2+^ levels. Thus, reduced expression of SERCA and MCU, implying lower Ca^2+^-buffering capacity of both organelles, may account for the higher cytosolic Ca^2+^ levels at rest that have been repeatedly reported in SOD1(G93A) paradigms (see above).

We may also speculate that different control of Ca^2+^ homeostasis and fluxes pertains to different cell types. In this respect, it is well recognized that SMs have a peculiar Ca^2+^ handling, being the ion deeply involved in the excitation-contraction coupling.

Finally, another important aspect to consider is the correlation between Ca^2+^ dysregulation and SOD1 aggregation since inclusions enriched in mutant variants of SOD1 are a well-established hallmark of SOD1-associated fALS forms [196,197]. In vitro studies reported that Ca^2+^ induces conformational changes on SOD1 fold by increasing its beta-sheet content and hydrophobicity, favoring its propensity to aggregate [198]. Similar results were obtained also in cultured MNs expressing SOD1 mutants, in which Ca^2+^ overload in the cytoplasm is not a consequence of SOD1 aggregation but rather causes the formation of SOD1 inclusions [199,200]. In addition, as in a vicious circle, SOD1 aggregates may promote a further increase in the levels of cytosolic Ca^2+^ [144]. On the other hand, an opposite effect is observed when Ca^2+^ buffering in cellular and animal models of fALS is increased, which resulted in a protective effect toward aggregation [201]. In this scenario, such data support the hypothesis that cytosolic Ca^2+^ overload contributes to ALS by favoring the formation of proteinaceous inclusions.

### 4.4. SOD1 Propagation in ALS

Another intriguing mechanism that may be involved in the pathogenesis of these forms of the disease is the fact that aggregated forms of mutated SOD1 can propagate from cell to cell and across different CNS regions in a way similar to prions and prionoids, the prototypes of such kind of events [202,203]. This process, that is common to other neurodegenerative diseases can take place through extracellular vesicles (EVs) of different sizes (exosomes or microvesicles) [204] and may contribute to the spread of disease and damage to vulnerable neurons in different CNS areas. Gomes and colleagues demonstrated, in the NSC34 MN-like cell line, that SOD1 aggregates propagate through EVs [205], while other studies showed that aggregated SOD1-containing EVs are mainly released from astrocytes, probably as a protective mechanism, but they may transfer their cargo to MNs causing their death [206]. EVs containing SOD1 aggregates have also been found in the plasma of SOD1-related familial ALS patients [207], as well as in the brain and spinal cord of SOD1(G93A) mice [208], further supporting their involvement in disease pathogenesis.

## 5. Conclusions

Starting from the 30-year-old seminal work involving SOD1 in ALS, it became more and more clear that mutant SOD1 is strictly involved in inheritable disease forms, leading to cytotoxicity and MN death. However, the mechanisms underlying the relationship between SOD1 and ALS are very complex and are not yet fully understood. We here reviewed how glial and skeletal muscle cells might play a central role in disease pathogenesis, as well as how alterations of Ca^2+^ dynamics and redox balancing might be involved in such dysfunctionalities (Figure 5). Original data demonstrating altered expression of Ca^2+^ transporters and disturbed local Ca^2+^ fluxes in primary cultures of SMs from hSOD1(G93A) Tg mice indicate that such alterations may be responsible for early SM dysfunctions in ALS, twisting the interplay between SMs and MNs and contributing to MN death. Further studies in adult/ill ALS mice and in human ALS patients will be necessary to further validate such a hypothesis.

## Figures and Tables

**Figure 1 antioxidants-11-00614-f001:**
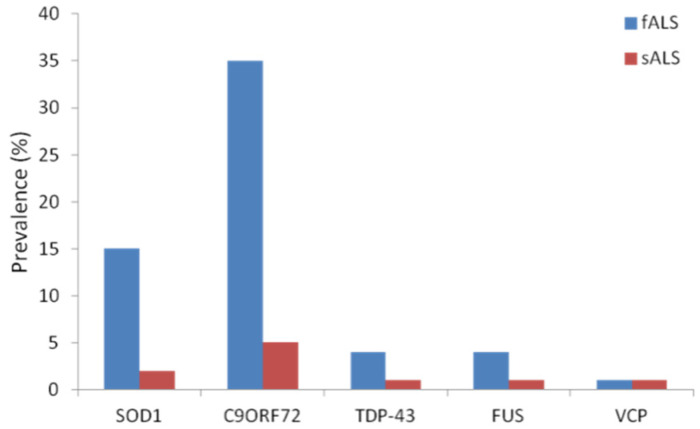
Prevalence (expressed as percentage) of mutations in the most important ALS-related genes in familial (fALS) or sporadic (sALS) cases, as reported in [6].

**Figure 2 antioxidants-11-00614-f002:**
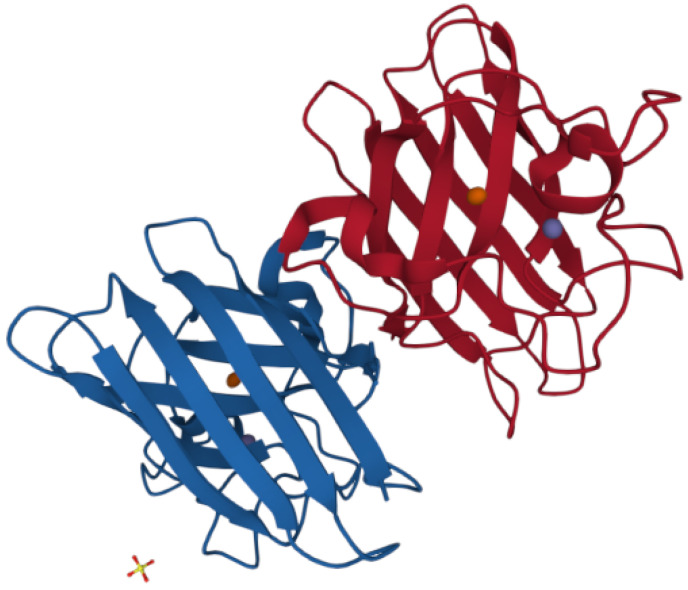
Structure of the dimeric human SOD1 (PDB 1PU0). It is possible to appreciate the β-barrel structure and the copper (orange) and the zinc (purple) atoms present in each subunit.

**Figure 3 antioxidants-11-00614-f003:**
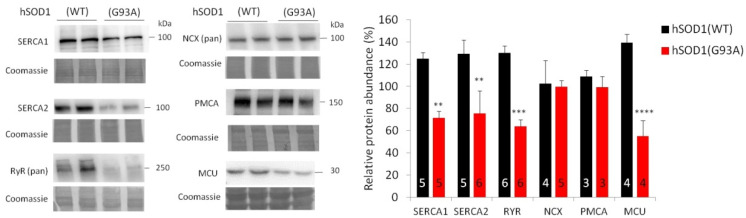
hSOD1(G93A) SMs express a lower amount of SR/ER and mitochondrial Ca^2+^ transporters compared to the control hSOD1(WT) counterpart. Primary differentiated myotubes (isolated from hSOD1(WT) and hSOD1(G93A) mice [150] (authorization N. 305/2017-PR, released on 6 April 2017)), cultured for 7 days as previously described [187] to induce in vitro differentiation to self-contracting myotubes, were lysed using an ice-cold buffer containing glycerol (10% (*w*/*v*)), SDS (2% (*w*/*v*)), Tris/HCl (62.5 mM, pH 6.8), and cocktails of protease and phosphatase inhibitors (Roche, Basel, Switzerland), and subjected to Western blot (WB) with antibodies to SERCA1 (Abcam, Cambridge, UK, cat. no. ab2819), SERCA2 (Santa Cruz Biotechnology, Dallas, TX, USA, cat. no. Sc-8095), RyR (Abcam, cat. no. ab2868), NCX (Cell Signaling, Danvers, MA, USA, cat. no. 79350), PMCA (Sigma, Tokyo, Japan, cat. no. P6363), and MCU (Sigma, cat. no. HPA016480). Representative WBs for each protein are shown in the left panel, while the bar diagram on the right reports the corresponding densitometric analyses. Total protein quantification, WB, and densitometric analyses were accomplished as described elsewhere [150]. Values are reported as mean ± standard error of the mean (SEM). Here and after, numbers inside the bars indicate the number of replicates. ** *p*-value < 0.01, *** *p*-value < 0.001, **** *p*-value < 0.0001; two-way ANOVA followed by Sidak’s multiple comparison.

**Figure 4 antioxidants-11-00614-f004:**
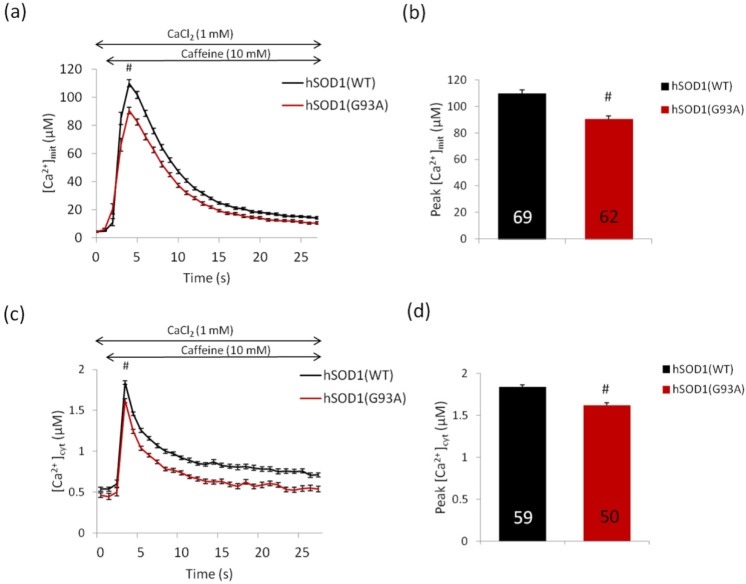
hSOD1(G93A) SMs display reduced mitochondrial and cytosolic Ca^2+^ fluxes in comparison with hSOD1(WT) controls after stimulation with caffeine. Two day cultured primary myocytes were transfected with plasmids coding for either mitAEQ (**a**,**b**) or cytAEQ (**c**,**d**) probes. Measurements of mitochondrial [Ca^2+^]_mit_ and cytosolic [Ca^2+^]_cyt_ transients were carried out as in [188,190], except for the use of a PerkinElmer EnVision plate reader as previously described [191]. (**a**,**c**) The average kinetics of [Ca^2+^]_mit_ and [Ca^2+^]_cyt_ transients, respectively, upon stimulation with caffeine (10 mM) at the indicated time point, while the corresponding peak values are reported in the bar diagrams of (**b**,**d**). Data are reported as mean ± SEM; # *p*-value < 0.001, Student’s *t*-test.

**Figure 5 antioxidants-11-00614-f005:**
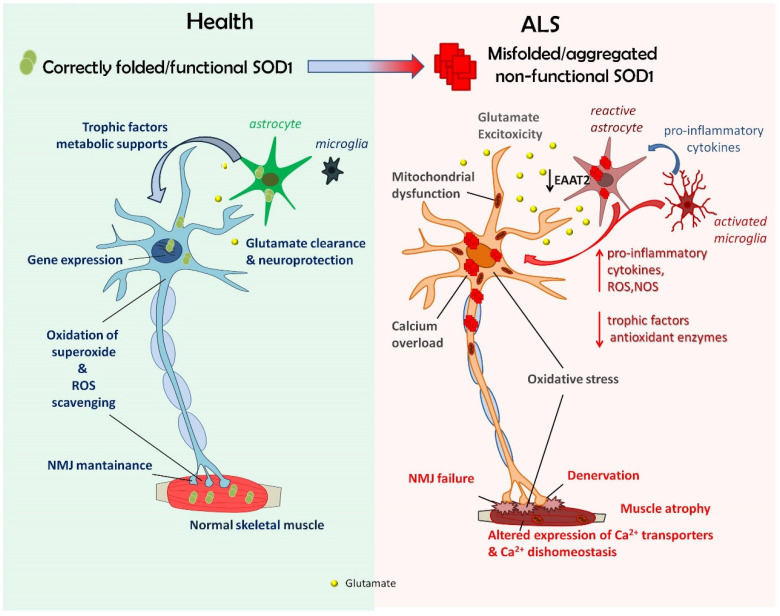
Irrespective of the mechanisms for SOD1-related ALS being the formation of cytotoxic polymeric/aggregated species of misfolded SOD1 and/or a loss of function of mutated/aggregated enzyme, alteration of SOD1 metabolism impinges on many cellular functions involving different cell types (i.e., motor neurons, glial cells, and skeletal myocytes) that might interact with each other to produce the pathologic phenotype.

## Data Availability

All of the data is contained within the article.

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
