# Peer review of "SOD1 in ALS: Taking Stock in Pathogenic Mechanisms and the Role of Glial and Muscle Cells"

_antioxidants, 2022, doi:10.3390/antiox11040614_

Round 1
Reviewer 1 Report
A nice, comprehensive review on the role of SOD1 and ALS mechanisms, particularly focused on non-MN-related mechanisms.
A couple small typos:
- Line 535 should read “Figure 3” not Figure 2
- Line 539 should read “Panels c and d” not panels b and c.
It is an interesting finding that mutant SOD1 can affect skeletal muscle Ca dynamics:
- The potential primacy of skeletal muscle pathology that leads to downstream NMJ disruption and MN death is well-summarized.
- One point of contradiction remains in the summarized papers and the presented data. The data presented suggest that, in mutant SOD1 myocytes, there is actually a decrease in both cytosolic and mitochondrial Ca2+. Perhaps suggesting Ca2+ remains sequestered in the sarcoplasmic reticulum?
- However, this seems to contradict later discussion (final 3 paragraphs beginning in line 556) that suggests elevated Ca is linked to ROS production and downstream NMJ and MN degeneration. Is there further clarification or published work on how the experimental data (reduced Ca) presented supports or refutes the reviewed literature (overabundance of Ca)?
Reviewer 2 Report
This review article describes the mechanistic role for SOD1 mutations in ALS. Overall, the paper is well-written and a good overview of the topic. Suggestions are made to improve the quality of the manuscript.
-Please specify the prevalence and incidence of ALS, distinguishing between fALS and sALS in different populations. Also provide an estimate for the relative prevalence of SOD1 mutations in ALS.
-Please mention other gene mutations that have also been associated with ALS (C9ORF72, TDP-43, etc) and their relative prevalence compared to SOD1 mutations.
-Brief mention of possible therapeutic avenues targeting SOD1 loss of function should be added.
-It is unusual to include unpublished primary data in a review article. Please clearly state within the manuscript and figure legends if Fig. 2-3 are unpublished data.
-Fig 2- a group analysis (2-way ANOVA) with multiple comparisons test should be performed here, rather than a t-test.
-There has been some evidence for EV-mediated SOD1 transfer in the context of ALS. Please briefly discuss in the manuscript.
-Any potential for targeting the longevity-associated genes (e.g., sirtuin family of proteins) in recovery of motor neuron degeneration in genetic animal models?
-Please correct all typos throughout (e.g., line 16, line 164 - misplaced punctuation).
